# A Novel Accelerometry-Based Metric to Improve Estimation of Whole-Body Mechanical Load

**DOI:** 10.3390/s21103398

**Published:** 2021-05-13

**Authors:** Enzo Hollville, Antoine Couturier, Gaël Guilhem, Giuseppe Rabita

**Affiliations:** French Institute of Sport (INSEP), Laboratory Sport, Expertise and Performance, EA7370 Paris, France; enzo.hollville@insep.fr (E.H.); antoine.couturier@insep.fr (A.C.); gael.guilhem@insep.fr (G.G.)

**Keywords:** Player Load, Accel’Rate, external load, team sport, training load monitoring, injury

## Abstract

While the Player Load is a widely-used parameter for physical demand quantification using wearable accelerometers, its calculation is subjected to potential errors related to rotational changes of the reference frame. The aims of this study were (i) to assess the concurrent validity of accelerometry-based Player Load against force plates; (ii) to validate a novel metric, the Accel’Rate overcoming this theoretical issue. Twenty-one recreational athlete males instrumented with two triaxial accelerometers positioned at the upper and lower back performed running-based locomotor movements at low and high intensity over six in-series force plates. We examined the validity of the Player Load and the Accel’Rate by using force plates. Standard error of the estimate was small to moderate for all tested conditions (Player Load: 0.45 to 0.87; Accel’Rate: 0.25 to 0.95). Accel’Rate displayed trivial to small mean biases (−1.0 to 6.1 a.u.) while the Player Load displayed systematic very large to extremely large mean biases (17.1 to 226.0 a.u.). These findings demonstrate a better concurrent validity of the Accel’Rate compared to the Player Load. This metric could be used to improve the estimation of whole-body mechanical load, easily accessible in sport training and competition settings.

## 1. Introduction

During the last few decades, the physical demand of team sports has greatly increased, as reflected by the increasing game frequency over a season, competition level and rules modifications [1]. Because the rise in physical demand has crucial implications for training periodization and injury prevention, its quantification and monitoring are essentials in various team sports [2,3,4,5]. In this context, the exponential development of wearable technology and load quantification algorithms provide access to various metrics that inform about internal and external load [6]. Although these lightweight devices are appealing, very few of these methods benefit from strong scientific evidence supporting their validity and reliability [7,8,9].

Among the wearable microsensors developed in the last few years, triaxial accelerometers provide one of the most significant advancements related to the quantification of physical demand [10]. In comparison to the GPS technology, these highly responsive inertial microsensors offer the substantial advantage of allowing the quantification of the physical demand elicited by various combinations of very fast movements in indoor and outdoor conditions [10,11,12]. To date, numerous accelerometry-based external load metrics have been proposed with the objective to provide an estimation of the whole-body mechanical load (i.e., external forces applied to the body/biomechanical loading experienced by the musculoskeletal system; [13]) [10,14,15]. These metrics mainly describe a cumulative or summative measure of accelerations and are valuable surrogates of segmental accelerations, and to a lower extent of the whole-body mechanical load [16]. Among them, the Player Load (Catapult, Victoria, Australia) is considered to be the most commonly-used metric for quantifying whole-body mechanical load in team sports [10,13,14,15,17,18]. During human locomotor tasks, the complex multi-dynamics of the body can be fairly estimated using force plates [19]. The use of ground reaction forces (GRF) measure has been used to infer about the external forces applied to the body (e.g., cumulative loading) during various locomotor tasks [12,16,20]. To be considered as a valid surrogate of the whole-body mechanical load, the Player Load and any accelerometry-based metrics must reflect the external load at the center-of-mass level using force plates [16]. Yet, no study has satisfactorily evaluated the validity of the Player Load in ecological conditions. To our knowledge, only one study assessed its concurrent validity in a laboratory setting [21]. In the study of Nicolella et al., nineteen accelerometers were simultaneously tested using an electrodynamic shaker table that imposed highly controlled oscillations to the accelerometers in three orthogonal directions. These authors questioned the validity of this metric based on the comparison between the Player Load given by the manufacturer (Catapult) and the Player Load calculated with its Cartesian formula:(1)$$Player\;Load=\sqrt{\left(a_{x_{i}}-a_{x_{i-1}}\right)²+\left(a_{y_{i}}-a_{y_{i-1}}\right)²+\left(a_{z_{i}}-a_{z_{i-1}}\right)²}$$
where $a_{x}$ is the mediolateral acceleration, $a_{y}$ the anteroposterior acceleration and $a_{z}$ the vertical acceleration. Nevertheless, beyond any potential sources that can influence the Player Load validity, the application of its Cartesian formula itself is questionable. As described above, the Player Load is defined as the square root of the sum of the squared instantaneous rate of change in acceleration of *x*, *y* and *z* axes [22]. Considering the wearable characteristics of the accelerometers, the reference frame in which the Player Load is computed is by definition non-inertial since subjected to translational and rotational changes. While the translational changes of the reference frame are intrinsically measured by the wearable device itself, the rotational changes of the reference frame may lead to erroneous Player Load calculation if Galilean transformations are not considered due to the generation of fictitious accelerations by the device orientation changes.

We posit that the measurement of the instantaneous changes in the modulus of the 3-axes acceleration vector represents a potential alternative to bypass this theoretical issue, leading to the computation of the Accel’Rate metric:(2)$$Accel^{\prime }Rate=\left|\sqrt{{\left(a_{x_{i}}\right)}^{2}+{\left(a_{y_{i}}\right)}^{2}+{\left(a_{z_{i}}\right)}^{2}}-\sqrt{{\left(a_{x_{i-1}}\right)}^{2}+{\left(a_{y_{i-1}}\right)}^{2}+{\left(a_{z_{i-1}}\right)}^{2}}\right|$$
where $a_{y}$ is the anteroposterior acceleration, $a_{x}$ is the mediolateral acceleration and $a_{z}$ is the vertical acceleration. In contrast to each component of an acceleration vector, the modulus does not depend on the reference frame, thus Galilean transformations are not required when considering the device rotation changes, preventing from potential accumulation of fictitious accelerations when estimating the whole-body mechanical load on the field.

Therefore, the primary aim of the present study was to assess the concurrent validity of both Player Load and Accel’Rate metrics. This concurrent validity was evaluated during typical team sport locomotor activities executed in ecological conditions. Additionally, we sought to determine the influence of: (i) locomotor task intensity and (ii) the sensor position [upper back vs. lower back], on both tested metrics validity.

## 2. Materials and Methods

### 2.1. Participants

Twenty-one recreational team-sport players (21 males; age: 29.5 ± 7.9 years; height: 179.2 ± 4.8 cm; body mass: 76.0 ± 5.6 kg; >5 years team-sport experience; >5 h training/week) volunteered to take part in this study. Participants were informed regarding the nature, aims and risks associated with the experimental procedure. This study was conformed to the standards of the Declaration of Helsinki.

### 2.2. Experimental Protocol

The experimental procedure is depicted in Figure 1A. After they received instructions about the experimental protocol, all participants performed a standardized warm-up (i.e., 10 min of jogging at free pace and dynamic stretching; 10 min of sprint-specific drills including athletic movements, changes of direction, short bursts of acceleration and deceleration and progressive sprints until maximal sprinting over 30–40 m). Then, each participant was asked to perform four types of typical team sport movements in a randomized order: general locomotor movements (GLM) consisting of back and forth forward, lateral, and backward locomotor tasks; a running start (RS) consisting of 4 to 5 initial steps of a run starting on the force plates (i.e., standing start); a run at constant velocity (RCV), consisting of a steady-state run over the force plates area. The running trial started ten or twenty meters before the first force plate depending on the locomotor task intensity; and a simulated one-on-one (1 vs. 1), that consisted in a 2-m acceleration followed by a double-step and a side-step cutting maneuver over the force plates. The participants performed these running-based locomotor tasks at both low and high intensity according to the following instructions: “pace your effort at a low intensity level” or “realize an all-out performance, with maximum involvement”.

### 2.3. Instrumentation

#### 2.3.1. Force Plates

Locomotor tasks were carried out over six individual force plates connected in series (KI 9067; Kistler, Winterthur, Switzerland; piezoelectric sensors; 1.2 × 0.6 m). The second force plate was turned by 90 degrees such that the total length of the force plates area was 6.6 m (for details, see [20]) (Figure 1B). This system allowed recording vertical, anteroposterior and medial-lateral components of the GRF at a sampling frequency of 1000 Hz.

#### 2.3.2. Triaxial Accelerometry

During the experimental protocol, participants were instrumented with two triaxial accelerometers (MinimaxX S4, Catapult) sampled at 100 Hz (mass: 67 g, size: 88 × 50 × 19 mm). The first device was placed in the center of the upper back and worn in a standard manufacturer harness (Catapult) designed to minimize artifact movements between the device and the body. The sensor was located ~5 cm lower than the base of the neck (7th cervical cervical). The second device was securely attached directly on the skin by a bi-adhesive support on the middle of the lower back over the horizontal axis determined by the posterior superior iliac crests. The sensors have been calibrated by the manufacturer and not specifically re-calibrated/validated for this study. A study using similar accelerometers (MinimaxX 2.0) previously reported acceptable within- and between-device reliability in the lab and on the field in both static and dynamic conditions using a hydraulic testing machine (coefficient of variation [CV] < 2%; [22]). In the study of Barrett et al. [17] using MinimaxX S4, they reported high test-retest reliability of the Player Load at both upper (CV: 5.9% and ICC: 0.93) and lower back (CV: 5.2% and ICC: 0.97). Other publications on more recent generation of accelerometers from the same brand/company reported similar observations regarding the accelerometer reliability and validity (0.4 < CV < 6.7%, in [23]; intraclass correlation coefficient 0.77–1.0, in [21]).

### 2.4. Data Processing

Accelerometer signals were exported using the manufacturer software Sprint 5.1.0.1 (Catapult) while GRF signals were exported using Bioware (Kistler). All analyses were then performed using custom written scripts (Origin 2017, OriginLab Corporation, Northampton, MA, USA). To synchronize force plate signals with accelerometer signals, we asked the participants to make three successive jumps at the beginning of each trial. Then, after GRF data were down sampled at 100 Hz, force plate and accelerometer signals were synchronized by cross-correlation analyses [24]. The three components of the participant’s center-of-mass acceleration were measured based on GRF via Newton’s Second Law of Motion as follows:(3)$$a_{x}=f_{x}/m$$
(4)$$a_{y}=f_{y}/m$$
(5)$$a_{z}=f_{z}/m-g$$
where *a* is the center-of-mass acceleration, *g* is the vertical acceleration due to gravity (9.81 m·s^2^), *f* is the force applied, obtained from force plates, and *m* is the mass of the participant. The *x*, *y*, and *z* axes are oriented in the medial-lateral, anteroposterior and vertical directions in the terrestrial reference frame of force plates, respectively (for details, see [19]). We then calculated Player Load (Equation (1)) and Accel’Rate metrics (Equation (2)) based on the three components of accelerations measured on force plates. These measures were considered as criterion measures.

Triaxial accelerometers were used to provide the direct measurement of the acceleration of the body part to which it was attached, also named segmental acceleration [16]. Medial-lateral, anteroposterior and vertical components of the segmental acceleration were expressed in the orthogonal coordinate system of the accelerometer.

As previously mentioned, we aimed to evaluate the concurrent validity of the Player Load and to assess the concurrent validity of a new computation, the Accel’Rate. First, we calculated Player Load using its classical equation (Equation (1)) based on the instantaneous acceleration variation of each component. The computation of Player Load provided by the manufacturer software comprises additional data filtering techniques [21]. In a pilot analysis, we retrieved these procedures including low-pass filtering (zero-phase shifting 3rd order Butterworth 1 Hz) and a scaling factor (×4). We thus applied the same signal processing after computed the Player Load using its cartesian formula in order to get the same values as the software output [21]. Second, we calculated the Accel’Rate, a novel metric to estimate the whole-body mechanical load. Its computation is based on the instantaneous variations of the magnitude of the modulus acceleration vector (Equation (2)). This calculation allows to bypass abovementioned theoretical issues related to the non-Galilean referential frame of accelerometers. Similar low-pass filter (zero-phase shifting 3rd order Butterworth 1 Hz) and scaling factor (×4) were applied to the Accel’Rate data. A typical graphical representation of the two computation methods measured with accelerometers (Player Load vs. Accel’Rate) compared to the Accel’Rate method derived from force plates (defined as criterion method) is depicted in Figure 1C.

### 2.5. Statistical Analysis

For Player Load and Accel’Rate metrics, the agreement between data obtained using accelerometers (practical measures) and force plates (criterion measure) was examined [25]. The level of concordance was estimated by the Bland and Altman graphical representation [26] with a 95% limit of agreements (mean bias, 95% LoA; mean difference ±1.96 SD). To interpret mean bias, we used the modified Cohen scale based on standardization of the mean bias (i.e., divided by the criterion standard deviation): <0.20, trivial; 0.2–0.6, small; 0.6–1.2, moderate; 1.2–2.0, large; 2.0–4.0, very large; >4.0, extremely large. The standard error of the estimate (SEE) (95% confidence intervals) was calculated and standardized for the purpose of interpretation. The SEEs were described as trivial if <0.2; small between 0.2 and 0.6; moderate between 0.6 and 1.2; large between 1.2 and 2.0; and very large if >2. Linear regressions were performed using Pearson’s r product-moment correlation.

## 3. Results

Concurrent validity of both Player Load and Accel’Rate are presented in Table 1, while Bland-Altman plots for graphical interpretation are depicted in Figure 2, Figure 3, Figure 4 and Figure 5. For the sake of clarity, we first present the data related to the upper back position and then the influence of sensor position on the metrics concurrent validity.

### 3.1. Concurrent Validity of the Player Load

For the Player Load metric, Pearson’s correlation coefficient showed a moderate to very large relationship (0.53 to 0.90) between the accelerometer (practical measure) and the force plates (criterion measure) depending on the type of locomotor task (Table 1). Specifically, large to very large correlations were found for GLM, RS and 1 vs. 1 conditions (0.68 to 0.90), whilst moderate to large correlations were found for the RCV condition (0.32 to 0.72). The SEEs were described as small to moderate for all tested conditions (0.45 to 0.87). Despite good correlation between Player Load and the criterion measure, we found systematic very large to extremely large mean biases (17.1 to 226.0 a.u.) of the Player load whatever the type of locomotor task and intensity (Table 1).

### 3.2. Concurrent Validity of the Accel’Rate

The Accel’Rate was calculated from the accelerometer data (practical measure) and compared against the Accel’Rate calculated from the force plates (criterion measure). Pearson’s correlation coefficients ranged from 0.71 to 0.97 (Table 1). We found large to nearly perfect relationships except for RCV, which presented a moderate linear dependence (0.31 to 0.65). The SEEs were described as small (0.25 to 0.60) except for RS at high intensity (0.73) and moderate for RCV at low and high intensity (0.78 to 0.95). The systematic mean biases observed with the Player Load calculation were largely reduced when using the Accel’Rate for every type of locomotor task (−1.0 to 6.1 a.u.; i.e., trivial to small) and was dependent on the task intensity and sensor unit location, as presented thereafter.

### 3.3. Effect of Locomotor Task Intensity

We observed an influence of the locomotor task intensity on the concurrent validity of the metrics, especially for Player Load. Mean biases (low vs. high intensity; 17.1 to 96.5 a.u. vs. 29.9 to 338.0 a.u.) and SEEs (0.45 to 0.97 vs. 0.51 to 0.97) were systematically larger, and correlation coefficients were lower at high intensity compared to low intensity for Player Load (0.34 to 0.90 vs. 0.32 to 0.87, respectively). Regarding Accel’Rate, mean biases were similar between low and high intensities (Table 1). Correlation coefficients (low vs. high intensity; 0.61 to 0.97 vs. 0.25 to 0.88) and SEEs (low vs. high intensity, 0.25 to 0.81 vs. 0.50 to 0.97) were respectively smaller and larger at high compared to low intensity.

### 3.4. Effect of Sensor Unit Location

At both upper and lower back locations, we observed large to very large correlations between Player Load and the criterion measure for GLM, RS and 1 vs. 1 (0.68 to 0.90). However, the upper back position showed better correlations for RCV at low and high intensity compared to the lower back position (upper back vs. lower back; 0.53 to 0.72 vs. 0.32 to 0.34). Similar SEEs data were also found for the two sensor locations using Player Load (Table 1). Each tested condition (i.e., locomotor task and intensity) showed very large to extremely large mean biases at upper and lower back positions using Player Load (17.1 to 338.0 a.u.). However, the upper back position systematically exhibited lower mean biases than the lower back position (17.1 to 226.0 a.u. vs. 39.7 to 338.0 a.u., respectively).

We observed less influence of sensor positions on the Accel’Rate concurrent validity. Specifically, we found a higher bias for the lower back in comparison with the upper back position (1.4 to 23.1 a.u. vs. −2.6 to 6.1 a.u., respectively). These mean biases were considered as trivial to small for the upper back position and as moderate to large for the lower back position except for 1 vs. 1 at high intensity (trivial bias). Moreover, the SEEs were larger in the lower back position compared to the upper back position except for RS at high intensity (Table 1).

## 4. Discussion

In this investigation, we sought to assess the concurrent validity of the Player Load metric, and to overcome theoretical issues related to its calculation. We therefore developed and quantified a novel metric, the Accel’Rate, with the ultimate goal of improving the estimation of the whole-body mechanical load using triaxial accelerometers. The present study has four novel findings: first, the Player Load inferred from the common accelerometry-based computation method did not match the whole-body mechanical load applied to the player’s center-of-mass in ecological conditions using in-series force plates. Second, the Accel’Rate measured with wearable accelerometers presented good to very good validity with the criterion measure (i.e., force plates) for various team-sport locomotor tasks executed at different intensities. Third, the locomotor task intensity substantially altered the accuracy and agreement of Player Load evaluation compared to Accel’Rate, which remained robust regardless of the task intensity. We posit that the novel Accel’Rate metric, based on instantaneous variations of the modulus of the 3-axes acceleration vector, is more suitable to reflect the whole-body mechanical load elicited by various locomotor tasks and intensity. Fourth, the use of classical accelerometer body-worn position between scapulae was shown to be a better alternative than trunk position (lumbar) during running-based locomotor tasks and may therefore be recommended during training and competitions.

### 4.1. Validity of the Player Load and Accel’Rate Metrics

To our knowledge, while previous studies aimed to quantify the validity of the Player Load (construct validity [27]; convergent validity [17]; concurrent validity in laboratory setting [21]), this study is the first to use a concurrent validity method in ecological conditions to assess this metric. A recent study quantified Player Load validity under highly controlled laboratory conditions [21]. However, in this protocol, the sensors were not subjected to the unavoidable rotations encountered in natural human movements. As aforementioned, these rotations should be the main source of errors of the Player Load calculation. Without any accurate correction of the rotation matrix, changes in instantaneous accelerations based on component vectors cannot be quantified [28]. Consequently, fictitious accelerations are generated by the device orientation changes if Galilean transformations are not considered. Accordingly, our results showed that Player Load consistently overestimated in very large proportions the criterion measures (e.g., 36 ± 9% to 68 ± 20% for the upper back position; Table 1) provided by the reference method (i.e., force plates). In line with previous findings, our results demonstrate that the Player Load does not accurately reflect the whole-body mechanical load estimated at the center-of-mass level [16]. Hence, the fictitious variations in acceleration components generated by the device orientation changes without considering the changes in reference frame between two consecutive samples are far from negligible. This is even more problematic when considering the extreme sensitivity of the Player Load to the locomotor task intensity with lower accuracy at high intensity (Figure 2A, Figure 3A, Figure 4A and Figure 5A, LI vs. HI). This finding is of crucial importance, considering that the accelerometer sensor tested in the present study is mainly used to appraise the whole-body mechanical load resulting from intense, unpredictable high-accelerative movements with rotations, as those executed in team sports [3,27,29,30,31,32,33].

Our results clearly show that the alternative use of the Accel’Rate could partly overcome the limitations associated with Player Load and in turn improve the estimation of the whole-body mechanical load as suggested by the good to very good concurrent validity obtained with the in-series force plates methodology (i.e., trivial to small mean biases at upper back position; Figure 2B, Figure 3B, Figure 4B and Figure 5B). Given that the Accel’Rate and the Player Load were computed from the same 3D acceleration signals, the difference in validity between both parameters seems to confirm the putative substantial influence of sensor rotation, when evaluating ecological locomotor tasks such as walking, running or specific cutting maneuvers executed in team sports. In a very recent study, other authors have attempted to correct the rate of changes in the acceleration component by determining the device orientation [28]. Using sensor fusion algorithms, they estimated the device absolute orientation in two consecutive steps to accurately describe the acceleration in all three movement planes. However, they noted that such procedure might be defeated for long period of recording, such as training or game play, due to the unavoidable signal drifting introduced by integrating the gyroscopes data.

### 4.2. Effect of Accelerometer Position

By definition, body-worn accelerometers measure the acceleration of the instrumented segment. Therefore, the whole-body mechanical load estimation is highly influenced by sensor unit position [16]. Contrary to our expectations, the upper back unit provides better concurrent validity than the lower back unit. Previously, Barrett et al. found an overestimation of about 16% of the Player Load assessed from the upper back position in comparison to the lower back position [17]. However, to evaluate the influence of the sensor unit position, these authors used the lower back position as the criterion measure for the center-of-mass accelerations and Player Load computation. Our findings inferred from gold standard in-series force plates showed that the lower back position presented systematic higher whole-body mechanical load values compared to the upper back position. Therefore, the classical upper back location (i.e., between scapulae) may be a better alternative to estimate whole-body mechanical load using triaxial accelerometers. The finding that the upper back sensor location resulted in a closer match to the GRF accelerations compared to the lower back location can originate from the fact that the upper and middle part of trunk and arm segments account for a large proportion of the total body mass (~40%, [34]). Therefore, upper back accelerometer may provide an estimate of this upper trunk motion, more likely to match the total center-of-mass motion. In addition, the complex rotational movements of the lumbo–pelvic–hip resulting from the investigated movements (e.g., internal/external rotation, flexion/extension, abduction/adduction) may have fictitiously adding noise to acceleration signals recorded at the lower back position [35].

### 4.3. Limitations

In this study, we tested male recreational team-sport athletes experienced in such motor tasks with likely lower strength and power capabilities than elite athletes. Therefore, our observations are not fully generalizable to a broader range of population including female individuals or elite athletes. It is likely that the same motor tasks executed by elite players will exacerbate the differences observed between the two metrics and the criterion methods at high intensity. While the variability observed between individuals in the present study suggests the concurrent validity of these metrics would be persist with female athletes, this remains to be tested. The bi-adhesive support used for the lower back position may not appear as the best fixation design. However, before the measurements, we did a pilot analysis where we compared a bi-adhesive support with the accelerometer directly attached on the skin (with the participant’ short covering the device) versus the same device fixated onto a belt worn. We noticed a major issue with using a belt because of pelvis motion moved up and down the belt, thereby eliciting artificial accelerations. Furthermore, even firmly attached, the belt moved from its original location throughout the protocol. Using the bi-adhesive support, the accelerometer undoubtedly held better. To prevent for large skin and soft tissues motions during high intensity actions, we covered the device with the elastic band of the participant’ shorts. Sweating effect on fixation was controlled during the protocol by checking the location and firmness of the adhesive double-tape support between trials. In exceptional cases where sweating affected the fixation, we removed the device, dried the lower back area and put the device back at the exact same location with new double-side adhesive tape. Previously, Barrett et al. (2014) found a 15.7 ± 9.7% lower Player Load at the upper back location compared to the lower back location (pouch fitted in a garment) when running on a treadmill. In our study, we observed similar results with higher Player Load at the lower back but with larger differences compared with Barrett et al. (2014) (e.g., 27–40% on average when running at constant velocity, Table 1). This may be in part attributable to the different type of fixations used in both studies. However, we are confident that this would not change our conclusion regarding the systematic larger biases found at the lower back position compared to the upper back.

### 4.4. Practical Applications

Based on the present findings, practitioners should be aware that the Player Load metric does not accurately estimate their athletes’ whole-body mechanical load. Therefore, users must be cautious when evaluating a player’s whole-body mechanical load, as a blind unaware use of this variable can lead to misinterpretations and inappropriate monitoring and management of the physical load. In contrast, the novel metric Accel’Rate may better reflects the whole-body mechanical load estimated at the center-of-mass level during low and high intensity locomotor tasks and can be easily implemented.

## 5. Conclusions and Perspectives

This study shows that the classical Player Load computation presents substantial limitations to be used as a surrogate to estimate the whole-body mechanical load. In contrast, the Accel’Rate, a novel metric based on the modulus of the 3-axes acceleration vector, presents a very good concurrent validity and is not influenced by locomotor task intensity. This suggests that Accel’Rate may be used as a surrogate of the whole-body mechanical load in training and competition settings in various sport activities. Combined with this parameter, the commonly used upper back position best matched the center-of-mass accelerations and may be the most appropriate location to use in team sports.

The Accel’Rate parameter may also have some interesting clinical applications to estimate the whole-body mechanical load that individuals face during daily activities. Among different perspectives, further studies may verify whether this parameter is able to quantify activities of daily living for the remote monitoring of patients with chronic diseases or whether it could improve energy expenditure predictions from triaxial accelerometers [36,37,38].

## 6. Patents

The authors (E.H., A.C., G.G., and G.R.) are co-inventors of the European Patent no. (FR18/4435592) which is owned by the French Institute of Sport (INSEP) and contains scientific content related to that presented in the manuscript. Priority Date: 2017/08/28; International Filing Date: 2018/08/28; International Publication Date: 2019/03/07.

## Figures and Tables

**Figure 1 sensors-21-03398-f001:**
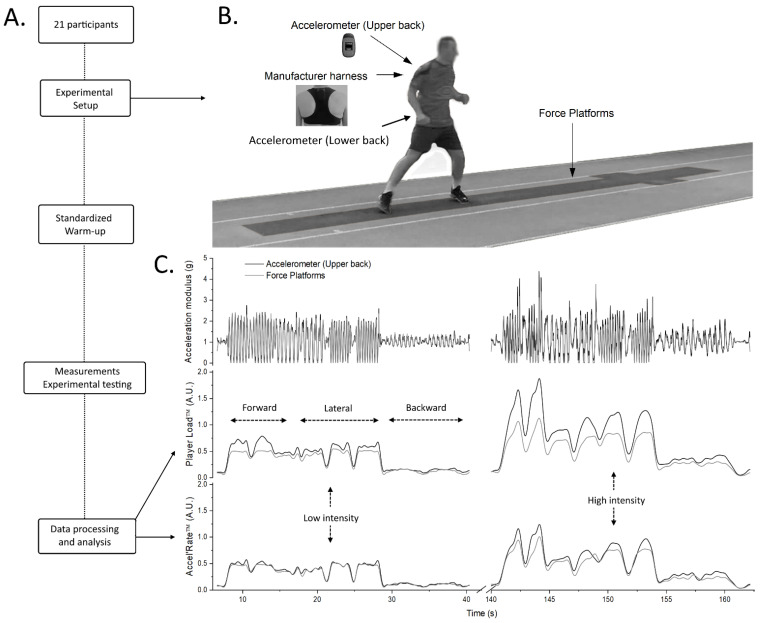
Overview of the experimental design, instrumentation setup and measurements from the concurrent validity protocol. (**A**): Experimental procedure. (**B**): Representation of the force plates area (total length of 6.6 m) and the position of the accelerometers. (**C**): Typical examples of acceleration modulus (top layer), Player Load (middle layer) and Accel’Rate (bottom layer) metrics during the general locomotor movements (GLM) at low (left) and high (right) intensity. Data were obtained from the in-series force plates (light grey traces) and the accelerometer positioned at the upper back (black traces; classical position used in team-sports).

**Figure 2 sensors-21-03398-f002:**
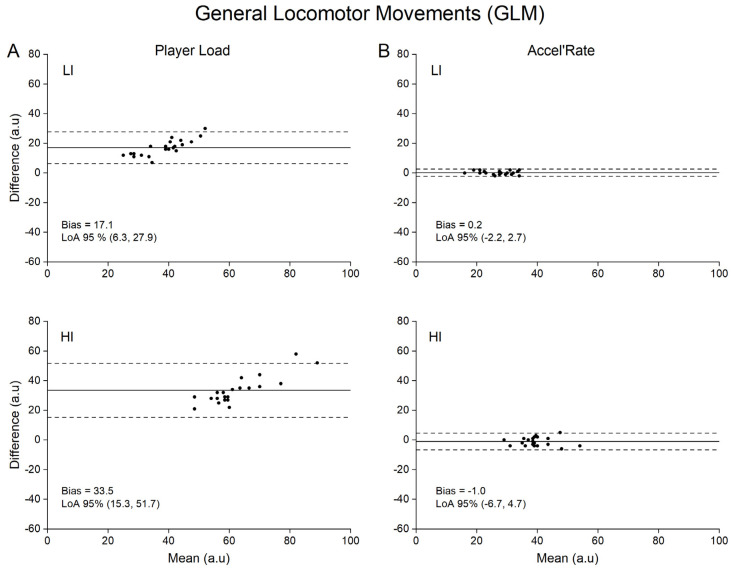
Bland-Altman plots of difference between criterion and practical measurements for general locomotor movements (GLM). For the sake of clarity, only data obtained for the upper back position are depicted. (**A**): Player Load measured by accelerometer (practical) versus by force plates (criterion) at low (LI) and high (HI) intensity; (**B**): Accel’Rate measured by accelerometer (practical) versus by force plates (criterion) at low (LI) and high (HI) intensity. Solid lines depict mean bias and dashed lines depict 95% limits of agreement (LoA).

**Figure 3 sensors-21-03398-f003:**
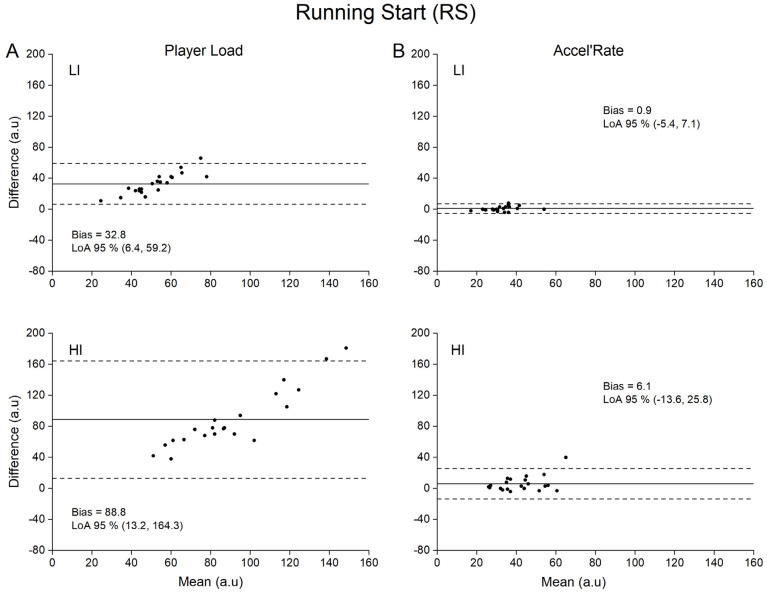
Bland-Altman plots of difference between criterion and practical measurements for running start (RS). For the sake of clarity, only data obtained for the upper back position are depicted. (**A**): Player Load measured by accelerometer (practical) versus by force plates (criterion) at low (LI) and high (HI) intensity; (**B**): Accel’Rate measured by accelerometer (practical) versus by force plates (criterion) at low (LI) and high (HI) intensity. Solid lines depict mean bias and dashed lines depict 95% limits of agreement (LoA).

**Figure 4 sensors-21-03398-f004:**
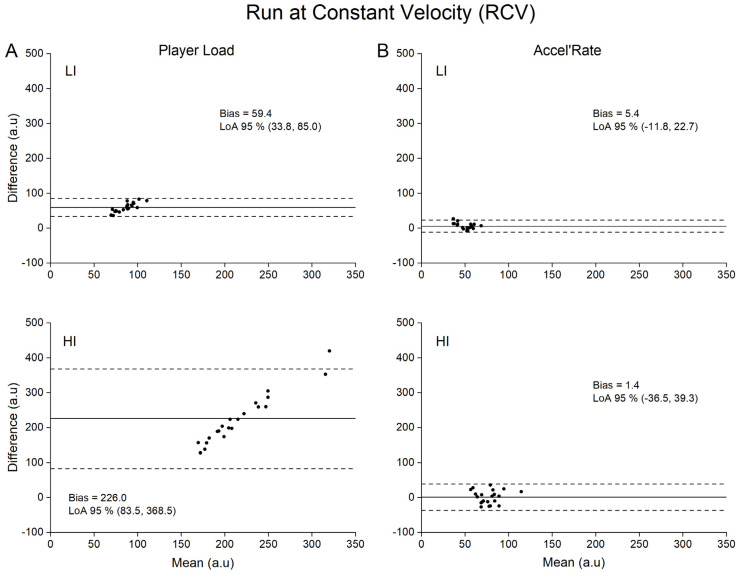
Bland-Altman plots of difference between criterion and practical measurements for run at constant velocity (RCV). For the sake of clarity, only data obtained for the upper back position are depicted. (**A**): Player Load measured by accelerometer (practical) versus by force plates (criterion) at low (LI) and high (HI) intensity; (**B**): Accel’Rate measured by accelerometer (practical) versus by force plates (criterion) at low (LI) and high (HI) intensity. Solid lines depict mean bias and dashed lines depict 95% limits of agreement (LoA).

**Figure 5 sensors-21-03398-f005:**
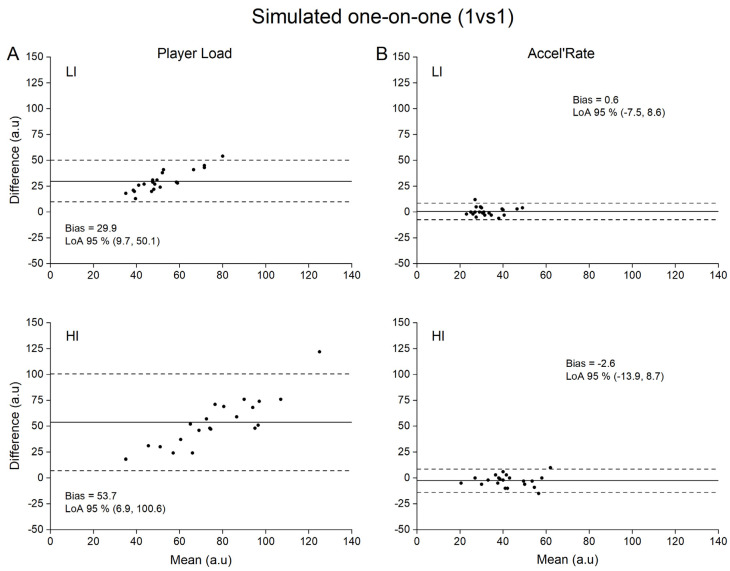
Bland-Altman plots of difference between criterion and practical measurements for simulated one-on-one (1 vs. 1). For the sake of clarity, only data obtained for the upper back position are depicted. (**A**): Player Load measured by accelerometer (practical) versus by force plates (criterion) at low (LI) and high (HI) intensity; (**B**): Accel’Rate measured by accelerometer (practical) versus by force plates (criterion) at low (LI) and high (HI) intensity. Solid lines depict mean bias and dashed lines depict 95% limits of agreement (LoA).

**Table 1 sensors-21-03398-t001:** Concurrent validity of the Player Load and the Accel’Rate measured with the triaxial accelerometers at the upper back and lower back positions and compared to the force plates gold standard method. Mean bias (in arbitrary units, a.u.) and standardized mean bias [±limits of agreements, LoA], Pearson correlation coefficient and standard error of estimate standardized (SEE) [confidence interval, 95% CI] were calculated for the 4-tested running-based locomotor task, at low and high intensity.

Movements and Intensity		Player Load				Accel’Rate			
		Upper Back		Lower Back		Upper Back		Lower Back	
General Locomotor movements (GLM)									
Low intensity	Pearson’s r	0.90	[0.77–0.96]	0.84	[0.63–0.93]	0.97	[0.93–0.99]	0.90	[0.76–0.96]
	Mean bias	17.10	(10.78)	39.66	(21.71)	0.24	(2.47)	4.10	(5.51)
	Standardized mean bias	3.08	(0.45)	7.16	(0.91)	0.05	(0.11)	0.79	(0.25)
	Standardized SEE	0.45	[0.34–0.65]	0.56	[0.43–0.82]	0.25	[0.19–0.36]	0.46	[0.35–0.67]
High intensity	Pearson r	0.87	[0.70–0.95]	0.80	[0.57–0.92]	0.88	[0.71–0.95]	0.72	[0.42–0.88]
	Mean bias	33.48	(18.21)	81.9	(34.78)	−1.00	(5.68)	4.81	(8.87)
	Standardized mean bias	5.06	(0.64)	12.38	(1.22)	−0.17	(0.23)	0.82	(0.35)
	Standardized SEE	0.51	[0.39–0.75]	0.61	[0.47–0.89]	0.50	[0.38–0.72]	0.71	[0.54–1.04]
Running Start (RS)									
Low intensity	Pearson’s r	0.82	[0.61–0.93]	0.81	[0.58–0.92]	0.92	[0.82–0.97]	0.91	[0.79–0.96]
	Mean bias	32.76	(26.38)	54.33	(39.02)	0.86	(6.24)	6.19	(10.70)
	Standardized mean bias	4.23	(0.79)	7.01	(1.17)	0.12	(0.20)	0.86	(0.34)
	Standardized SEE	0.58	[0.44–0.85]	0.60	[0.46–0.88]	0.39	[0.30–0.58]	0.43	[0.32–0.62]
High intensity	Pearson r	0.68	[0.35–0.86]	0.71	[0.41–0.88]	0.71	[0.39–0.87]	0.80	[0.57–0.92]
	Mean bias	88.76	(75.55)	121.19	(77.61)	6.09	(19.74)	6.90	(16.07)
	Standardized mean bias	7.48	(1.48)	10.21	(1.52)	0.59	(0.44)	0.67	(0.36)
	Standardized SEE	0.75	[0.57–1.10]	0.72	[0.55–1.05]	0.73	[0.55–1.06]	0.61	[0.47–0.90]
Run at Constant Velocity (RCV)									
Low intensity	Pearson’s r	0.72	[0.41–0.88]	0.34	[−0.11–0.67]	0.65	[0.30–0.84]	0.61	[0.25–0.83]
	Mean bias	59.38	(25.6)	96.52	(45.33)	5.43	(17.25)	14.57	(18.83)
	Standardized mean bias	9.01	(0.9)	14.65	(1.6)	0.47	(0.35)	1.26	(0.38)
	Standardized SEE	0.72	[0.55–1.05]	0.97	[0.73–1.41]	0.78	[0.59–1.14]	0.81	[0.62–1.18]
High intensity	Pearson’s r	0.53	[0.13–0.78]	0.32	[−0.13–0.66]	0.31	[−0.14–0.66]	0.25	[−0.20–0.62]
	Mean bias	226	(142.50)	337.96	(136.01)	1.38	(37.87)	23.14	(47.22)
	Standardized mean bias	21.88	(3.20)	32.71	(3.06)	0.09	(0.55)	1.44	(0.68)
	Standardized SEE	0.87	[0.66–1.27]	0.97	[0.74–1.42]	0.95	[0.74–1.42]	0.97	[0.75–1.45]
Simulated one-on-one (1 vs. 1)									
Low intensity	Pearson’s r	0.89	[0.75–0.96]	0.87	[0.71–0.95]	0.84	[0.65–0.94]	0.75	[0.48–0.89]
	Mean bias	29.91	(20.24)	60.76	(31.91)	0.57	(8.07)	5.71	(11.66)
	Standardized mean bias	3.75	(0.59)	7.62	(0.93)	0.08	(0.26)	0.78	(0.38)
	Standardized SEE	0.46	[0.35–0.67]	0.50	[0.49–0.93]	0.55	[0.42–0.80]	0.68	[0.51–0.99]
High intensity	Pearson’s r	0.78	[0.53–0.91]	0.85	[0.67–0.94]	0.86	[0.69–0.94]	0.84	[0.63–0.93]
	Mean bias	53.71	(46.84)	101.57	(55.86)	−2.62	(11.30)	1.43	(13.19)
	Standardized mean bias	4.22	(0.85)	7.98	(1.02)	−0.24	(0.24)	0.13	(0.28)
	Standardized SEE	0.64	[0.16–0.40]	0.53	[0.16–0.40]	0.52	[0.39–0.76]	0.56	[0.43–0.82]

## Data Availability

The data presented in this study are available on request from the corresponding author.

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
