# Peer review of "A Novel Accelerometry-Based Metric to Improve Estimation of Whole-Body Mechanical Load"

_sensors, 2021, doi:10.3390/s21103398_

Round 1
Reviewer 1 Report
This is an empirical study aimed to test the reliability and measure errors with a new accelerometer-based calculation. I am delight to see the authors conducted a well-design research to show the concurrent validity of the novel metric “AccelRate” with traditional metric “Player load”. I believe this study conducted with sound methodologies and well interpretation of research outcomes. In my opinion, this is a high quality of paper and is suitable to be published in Sensor. I have only few minor suggestions to authors.
1) Your study recruited 21 recreational athletes. Their physical profiles and training status should be reported.
2) Description of “a standardized warm-up” is insufficient. Please be detailed.
3) It is better to provide an illustration for experimental procedure.
4) I am curious how the authors ensure the stabilization of lower back device. I think bi-adhesive support is questionable, particular during sweating.
5) The x axe of force plate should be medial-lateral direction.
6) Line 194, based on Pearson correlation analysis, “prediction” is inappropriate.
7) Table 1, it is incredible to see large gap between player load and AccelRate in measuring Run at Constant Velocity.
Reviewer 2 Report
Introduction
it is necessary to add more information about the state of the art: for example on Accel’Rate parameter (a novel metric based on the modulus of the 3-axes acceleration vector)
Materials and Methods
L87:
The sample analyzed is not representative and limited, for example: why did you select only males?
L112
Have the sensors used (MinimaxX S4, Catapult, Victoria, Australia) been previously validated in a Gait and Motion Analysis Laboratory?
L 116:
why was a bi-adhesive used and not a belt to fix the sensor? How does sensor attachment affect the results?
Round 2
Reviewer 2 Report
Authors (E.H., A.C., G.G., and G.R.) are co-inventors of the European Patent no. (FR18/4435592) which contains scientific content (computational methods of Accel’Rate calculation) and this is the first publication using this novel metric useful to evaluate its validity.
Authors declare no conflict of interest, I think it is more correct to modify this part by adding these details previously omitted
Author Response
Thank you for this remark. These details were presented in the dedicated 'Patent' section of the template proposed by Sensors.
Following reviewer's suggestion, we now have added these details in the section 'Conflict of interest':
"Conflicts of Interest: The authors (E.H., A.C., G.G., and G.R.) are co-inventors of the European Patent no. (FR18/4435592) which is owned by the French Institute of Sport (INSEP) and contains scientific content related to that presented in the manuscript. Priority Date: 2017/08/28; International Filing Date: 2018/08/28; International Publication Date: 2019/03/07."